# Nutrient Intake and Dietary Acid Load of Special Diets in the NHANES: A Descriptive Analysis (2009–2018)

**DOI:** 10.3390/ijerph19095748

**Published:** 2022-05-09

**Authors:** Maximilian Andreas Storz, Alexander Müller, Alvaro Luis Ronco

**Affiliations:** 1Department of Internal Medicine II, Center for Complementary Medicine, Freiburg University Hospital, Faculty of Medicine, University of Freiburg, 79106 Freiburg, Germany; alexander.mueller@uniklinik-freiburg.de; 2Unit of Oncology and Radiotherapy, Pereira Rossell Women’s Hospital, Bvard. Artigas 1590, Montevideo 11600, Uruguay; alv.ronco58@gmail.com

**Keywords:** dietary acid load, PRAL, NEAP, plant-based, special diet, diabetic, weight-loss, low-fat

## Abstract

Western diets are characterized by a high dietary acid load (DAL), which has been associated with adverse clinical outcomes, including type-2-diabetes and metabolic syndrome. Effective dietary strategies to lower DAL are urgently warranted. Plant-based diets (PBD), including vegetarian and vegan diets, are an effective measure to reduce DAL. Notably, not every individual wishes to adopt a PBD. Instead, many people rely on special diets promising comparable health benefits. The effects of those diets on DAL have rarely been investigated. Using data from the National Health and Nutrition Examination Surveys, we aimed to quantify DAL and nutrient intake in the most popular special diets in the United States, including weight-loss, low fat, low salt, low sugar, and diabetic diets. Our analysis included 3552 individuals on a special diet. The most popular diets were the weight-loss diet (*n* = 1886 individuals) and the diabetic diet (*n* = 728). Energy intake was below 2000 kcal/d for all diets; however, there were no statistically significant intergroup differences. DAL scores were positive for all special diets (>9 mEq/d), suggesting acidifying properties. Acid load scores of special diets did not differ significantly and were comparable to a standard Western diet. None of the examined diets was associated with alkaline properties.

## 1. Introduction

Western diets are characterized by an excessive intake of highly processed and refined foods as well as high contents of added sugars, salt, and (saturated) fat and protein from meat and other animal products [1]. These contemporary diets in industrialized nations are largely acid-inducing and characterized by a high Dietary Acid Load (DAL) [2]. Meat and dairy products are abundant in sulfur-containing amino acids (cysteine, homocysteine, and methionine) and preservative phosphate, which have acidifying effects on the human body [3,4].

Sulfur-containing amino acids are catabolized to sulfate, a non-metabolizable anion and a major contributor to DAL [3,5]. Foods that are abundant in phosphate may also supply acid equivalents, depending on the cation that is attached to the phosphate anion [6]. Phosphoric acid (H_3_PO_4_) in dairy products and certain soft drinks, as well as other polyphosphates, may acidify urine [7].

At the same time, Western diets are deficient in alkalizing plant foods, such as vegetables, fruits, legumes, various nuts, and seeds [8]. The insufficient intake of plant foods is associated with a reduced intake of potassium salts of metabolizable organic anions (malate and citrate) [9]. The latter undergo combustion in the human body to yield bicarbonate, which can consume hydrogen ions when metabolized and thus have alkalizing effects [2,3,9].

An imbalance in alkalizing and acidifying foods may result in a high DAL, which has been associated with numerous adverse health outcomes, including type-2-diabetes, hyperuricemia, and metabolic syndrome [6,10,11]. Furthermore, elevated DAL scores have been associated with systemic inflammation and markers of cardiovascular disease [12,13].

One of the major underlying pathomechanisms is the induction of low-grade metabolic acidosis as a consequence of a persistent acidogenic diet, which raises the likelihood of an increased hydrogen ion surplus and chronically lower levels of serum bicarbonate [14,15]. This may particularly be the case when compensatory processes become less efficient and remain unresolved by sustainable dietary adjustments of bicarbonate [14]. Low-grade metabolic acidosis has been associated with hypercortisolism, which decreases insulin sensitivity and increases insulin resistance [5,14,16]. High DAL and subsequent metabolic acidosis have also been linked to decreased adiponectin levels [17], a hormone that enhances insulin sensitivity and possesses anti-inflammatory properties [18]. In metabolic acidosis, circulating adiponectin is lowered through inhibition of adiponectin gene transcription in adipocytes, which may, in turn, also decrease insulin sensitivity. While a more detailed review of additional health repercussions due to latent metabolic acidosis is beyond the scope of this paper, it is clear that a high DAL may adversely affect human health in the long run when not properly addressed.

As such, strategies to lower DAL are urgently warranted and currently subject to intensive clinical and pharmacological research [15].

Plant-based diets (particularly lacto-ovo-vegetarian and vegan diets) are an effective and potent measure to reduce DAL in clinical practice [9,19,20]. As reported by Müller et al., a 4-week plant-based (isocaloric vegan) dietary intervention significantly reduced DAL in healthy individuals [9]. Deriemaeker et al. reported comparable results investigating a lacto-ovo-vegetarian diet [20].

However, not everyone wishes to adopt a plant-based diet (PBD) [21]. Barriers to vegetarian diets, for example, include meat enjoyment, rigid eating routines, convenience, and difficulties in preparing vegetarian foods [22,23]. Instead, many people rely on special diets that promise comparable health benefits, including high-protein diets or low-sugar diets.

How well these special diets fare with regard to DAL has rarely been investigated. The present study sought to address this gap in the literature. Using data from the National Health and Nutrition Examination Surveys (NHANES) [24], this study aimed to quantify DAL and nutrient intake in some of the most popular special diets in the United States. Moreover, we contrasted the results to the DAL-lowering effects of vegetarian and vegan diets examined in previous studies.

## 2. Materials and Methods

### 2.1. The NHANES

The NHANES is a national program of studies designed to assess the health and nutritional status of adults and children in the United States of America [24]. It is a cross-sectional and nationally representative survey of the non-institutionalized U.S. civilian population and is characterized by a complex, multistage, probability sampling design [24,25]. The NHANES is an ongoing program and one of the largest of its kind. Each year, the survey examines a sample of about 5000 people. Participants are located in counties across the United States, 15 of which are visited each year. The two major components of the NHANES include the interview and the examination component. The interview includes questions covering demographic, socioeconomic, dietary, and health-related data. The examination component includes medical and physiological measurements and laboratory tests. All NHAHES participants visited the physician, and the examination also included laboratory tests administered by highly trained medical personnel.

The interview component is conducted in participants’ homes. The examination component is conducted in specially designed and equipped mobile examination centers. The study teams include a physician, a dentist, health and medical technicians, and dietary and health interviewers. Many of the staff involved are bilingual (English/Spanish). Additional background information on the NHANES is available on their homepage and from the overview brochure [26]. Data from the NHANES is often used for nutrition-related research, and key features of the NHANES have been described elsewhere in great detail [24]. NHANES was approved by the National Centre for Health Statistics research ethics review board, and informed consent was obtained for all participants.

### 2.2. The NHANES Dietary Interview

The NHANES includes a large dietary interview module designed to obtain detailed dietary intake information from participants [27,28]. The mainstay of this module is a 24-h recall administered by specifically trained interviewers to estimate energy and nutrient intake for all participants. The dietary interview component called “What We Eat in America”, is conducted as a partnership between the U.S. Department of Agriculture and the U.S. Department of Health and Human Services. The National Center for Health Statistics is responsible for the survey sample design and all aspects of data collection. The Food Surveys Research Group of the U.S. Department of Agriculture is responsible for the dietary data collection methodology, maintenance of the databases used to code and process the data, and data review and processing.

Details on the specific dietary data collection methods in the NHANES, such as the Automated Multiple-Pass Method (AMPM), may be obtained from the official homepage. The strengths and limitations of the NHANES dietary data have been discussed elsewhere in detail [27].

A recent study using NHANES data demonstrated that more than 17% of U.S. adults aged 20 years or older were on a special diet on a given day between 2015–2018 [29]. Stierman et al. reported that the most common types of special diets reported among all adults during the aforementioned timeframe were weight loss or low-calorie diets [29]. Based on the reliability of the NHANES dietary module and the included questions on special diets (see below), it was deemed appropriate for an exploration of DAL in those particular diets.

### 2.3. Special Diets in the NHANES

The dietary module included several questions on special diets. Participants were asked, “Are you currently on any kind of diet, either to lose weight or for some other health-related reason?” Those answering with “yes” were then asked, “What kind of diet are you on?”. No list of diets or standardized definitions were provided as part of the interview. Instead, responses were categorized by specially trained interviewers to the following dietary patterns: “weight loss or low calorie diets”, “low fat/low cholesterol diet”, “low salt/low sodium diet”, “sugar free/low sugar diet”, “low fiber diet”, “high fiber diet”, “diabetic diet”, “low carbohydrate diet”, “weight gain/muscle building diet”, “high protein diet”, or “other special diet”. For the lack of clear definitions of those diets in the NHANES, we performed a descriptive analysis of total energy intake, macronutrient distribution and nutrient intake for each of the aforementioned special diets. Using this approach, we sought to better characterize the diets with regard to the nutritional factors that influence DAL.

We sought to quantify the DAL of those special diets and appended 5 NHANES cycles (2009/2010, 2011/2012, 2013/2014, 2015/2016, and 2017/2018). In light of the available case numbers after a first exploratory analysis, the DAL scores of the following special diets were estimated: (1) weight-loss/low-calorie diet, (2) low-fat/low-cholesterol diet, (3) low-salt/low-sodium diet, (4) sugar-free/low-sugar diet, and (5) diabetic diet. The gluten-free or celiac diet and the renal or kidney diet that were added from 2009–2010, onwards were not considered for this analysis due to the small case numbers available. Only individuals aged 18 years or older were included in the present analysis.

### 2.4. Dietary Acid Load Estimations

We used 2 widely established formulas to calculate DAL from daily nutrient intake [30,31]. The employed methods have been discussed elsewhere in great detail [9]. In brief, we estimated the Potential Renal Acid Load (PRAL) from a diet based on a validated formula by Remer et al. that considers ionic dissociation, sulfur metabolism, and intestinal absorption rates for the following micro and macronutrients: potassium, phosphate, magnesium, calcium, and protein [9,30]. PRAL_R_ was calculated as follows:PRAL (mEq/day) = (0.49 × total protein (g/day)) + (0.037 × phosphorus (mg/day)) − (0.021 × potassium (mg/day)) − (0.026 × magnesium (mg/day)) − (0.013 × calcium (mg/day))(1)

Net endogenous acid production (NEAP) was estimated based on a formula proposed by Frassetto et al., considering the dietary content of potassium and protein intake [5,31]. NEAP_F_ was calculated as follows:NEAP_F_ (mEq/d) = (54.4 × protein (g/d)/potassium (mEq/d)) − 10.2 (2)

### 2.5. Statistical Analysis

For the statistical analysis, we used Stata version 14 (StataCorp., College Stadion, TX, USA) and constructed appropriate sample weights to account for the complex, multistage, probability sampling design of the NHANES. We described continuous variables with their mean and the corresponding standard error in parenthesis. Categorical variables were described in the following format: number of observations (weighted proportions (standard error)). This way of data presentation is in full accordance with the data presentation standards for proportions by the National Center for Health Statistics (NCHS). Unreliable proportions were explicitly marked with superscript letters [32]. Standard errors were estimated using Taylor series linearization to account for the complex sampling design. Stata’s design-adjusted Rao–Scott test and multivariate linear regression analyses (followed by adjusted Wald tests and Stata’s margins function) were used to test for potential intergroup differences between special diets. Marginsplots were used to graph statistics from fitted models. All tests were two-sided and statistical significance was determined at α = 0.05.

## 3. Results

Our analysis included *n* = 3552 individuals (unweighted) aged 18 years or older on a special diet with a complete dataset, which may be extrapolated to represent *n* = 32,194,273 individuals in the US. The most popular special diets in this sample were the weight-loss diet (followed by *n* = 1886 individuals (unweighted)) and the diabetic diet (*n* = 728 (unweighted)). Consistent with the previous literature on special diets in the NHANES, more women were on a special diet than men (Table 1) [29].

Additional sample characteristics and total energy and nutrient intake are presented in Table 1. The mean age varied significantly between groups (Table 1). Individuals on a diabetic diet were, on average, significantly older (60.90 years) than individuals on a weight loss diet (45.91 years). Statistically significant differences were also found with regard to the weighted proportions of race/ethnicity (Table 1).

Energy intake was below 2000 kcal/d for all examined diets, and there were no statistically significant intergroup differences (Table 1). With 1821.81 kcal/d, individuals on a diabetic diet having the lowest energy intake.

Diets differed significantly with regard to macronutrient distribution (and energy derived from macronutrients) but not with regard to total protein intake, which is relevant for DAL estimations. Total fat intake was highest in the low-sugar diet group (79.40 g/d), whereas this group demonstrated the lowest total carbohydrate intake (approximately 190 g/d). The highest carbohydrate intake (241.69 g/d) was found in individuals reporting a low salt diet. No significant intergroup differences were found with regard to the minerals that play a role in DAL estimation. Potassium intake, in particular, was comparable across diets, ranging from 2625.65 (in individuals on a weight-loss diet) to 2742.61 mg/d (in individuals reporting a low-salt diet).

Based on the nutrient intake data presented in Table 1, we calculated mean DAL scores using the formulas presented in Section 2.4.

The mean PRAL_R_ scores and NEAP_F_ scores are shown in Table 2. Notably, all special diets were associated with PRAL_R_ scores > 9 mEq/d. There were no significant intergroup differences with regard to PRAL_R_ and NEAP_F_ scores (*p* = 0.328 and *p* = 0.052, respectively). The lowest PRAL_R_ scores were found in individuals reporting a low-salt diet, whereas the highest scores were found in participants reporting a low-sugar diet. A comparable picture was found for NEAP_F_ scores.

Multivariate linear regression was used to estimate DAL scores of special diets after adjustment for various confounders (age, sex, race, total energy intake). Significant regression equations were found (F(11,68) = 35.14 for PRAL_R_ and F(11,68) = 16.03 for NEAP_R_, *p* < 0.001 for both), with R^2^ values of 0.176 and 0.073, respectively. The results are shown in Table 3.

Marginsplots were used to graph statistics from the fitted models shown above. Figure 1 displays the marginal predicted values of PRAL_R_ (left panel) and NEAP_F_ (right panel) for each special diet at all possible increments of 10 units in age. The DAL scores were consistently positive for all special diets (above 0 mEq/d), which is indicative of acidifying properties. None of the special diets was associated with PRAL_R_ scores < 0 mEq/d, which indicates an alkaline character. As shown in Figure 1, there was a tendency of decreasing DAL scores with increasing age.

## 4. Discussion

A high DAL has been associated with numerous health repercussions and adverse clinical outcomes, including type 2 diabetes, cardiovascular disease, and certain cancers [13,33,34,35]. Elevated DAL scores were repeatedly associated with low-grade metabolic acidosis, which, in turn, is linked to hypercortisolism and insulin resistance [5,14]. Thus, effective dietary strategies to lower DAL are urgently warranted.

Using data from the NHANES (2007–2010), we previously demonstrated that a lacto-ovo vegetarian diet was associated with a reduction in DAL compared to the general population [36]. Median DAL scores in NHANES vegetarians were as follows: PRAL_R_ −0.44 mEq/d and NEAP_F_: 39.60 mEq/d, whereby the negative PRAL_R_ scores suggested a slightly alkaline diet. In the general non-vegetarian population denying a special diet, we observed a different picture. DAL scores were positive and as follows: PRAL_R_: 11.90 mEq/d and NEAP_F_: 53.59 mEq/d.

In accordance with previous studies, our data suggested that a vegetarian diet might be a suitable tool to reduce DAL scores [6,9,20,36]. However, not every individual wishes to change their dietary habits and adopt a plant-based (vegetarian or vegan) diet [37]. Often cited barriers to these diets include meat enjoyment, rigid eating routines, and difficulties in preparing vegetarian foods [22,23]. Instead, individuals often rely on special diets that promise comparable health benefits, including high-protein diets or low-sugar diets. In light of the high DAL of a typical Western diet (ranging between approximately 50 and 75 mEq/d [38,39]), the present study sought to explore the DAL of potential alternative special diets. The present study is thus an extension of our previous investigation of DAL in United States-based vegetarians [36].

All examined special diets were associated with positive DAL scores (above 0 mEq/d), indicating acidifying properties. None of the special diets was associated with an alkaline character (Figure 1). Unlike plant-based diets, such as the vegetarian and vegan diets, which often yield negative DAL scores [6,9,20], none of the examined diets in this study yielded PRAL_R_ scores <0 mEq/d.

Moreover, DAL scores of all examined special diets (weight loss, low fat, low salt, low sugar, and diabetic diet) were comparable to a standard Western diet in the NHANES [36]. Mean PRAL_R_ scores were 12.98 mEq/d for the weight-loss diet, 11.21 mEq/d for the low-fat diet, 9.37 mEq/d for the low-salt diet, 15.26 mEq/d for the low-sugar diet, and 12.17 mEq/d for the diabetic diet. As such, they were all positive and somewhat similar to the mean PRAL_R_ scores found in the NHANES general population (11.90 mEq/d) in our previous study [36].

Questions arise as to why this is the case, and it is thus important to have a detailed look at the nutritional factors in the examined special diets, which play a pivotal role in DAL calculations: total protein intake, potassium intake, phosphorus intake, calcium intake, and magnesium intake.

Total protein intake in special diets ranged from 77.37 g/d in individuals reporting a diabetic diet to 84.60 g/d in individuals reporting a low sugar diet (Table 1). In comparison to NHANES vegetarians, who had a mean intake of approximately 62 g/d [36], protein intake was substantially higher in all special diets. When glancing at the DAL calculation methods (Section 2.4), this could be one of the major factors explaining the higher DAL scores in special diets. The relatively lower protein intake (and subsequently reduced consumption of sulfur-containing amino acids) has been discussed earlier as one of the main factors why individuals on a vegetarian or vegan diet have lower DAL scores [36]. Notably, protein intake in all special diets exceeded the intake recommendations in the current Dietary Guidelines for Americans (2020–2025) [40].

Another factor worth discussing is phosphorus intake in special diets, which ranges from 1307.94 to 1410.43 mg/d. Compared to vegetarian and vegan diets, this intake is quite high. NHANES vegetarians, who frequently consumed phosphor-containing dairy products, had an average intake of 1162.08 mg of phosphorus per day [36]. NHANES special diets (in particular the low-sugar diet) exceeded this value. This may also explain why PRAL_R_ scores in special diets are higher than, for example, in NHANES vegetarians. In this context, it is also crucial to take the weighting factor of 0.037 into account (see Section 2.4).

Another important aspect to consider is potassium intake. Potassium intake in special diets ranged from 2625.65 mg/d in weight loss diets to 2742.61 mg/d in low salt diets. While essentially above the recommended values for females in the current Dietary Guidelines for Americans (2020–2025) [40], these amounts are well below the intake in vegetarian and vegan diets. An intake of 4000 or 5000 mg/d of potassium is not uncommon in vegan diets [41] and may contribute to reduced acid load from the diet, given that potassium salts of metabolizable organic anions (malate and citrate) have alkalizing effects [2,3,9]. The comparably low potassium intake in all special diets in our sample may thus have also contributed to the estimated DAL scores.

Finally, it is worth taking a quick glance at macronutrient distribution in special diets. The current Dietary Guidelines for Americans (2020–2025) recommend an intake of at least 45% of total energy from carbohydrates [40]. Two of the special diets, namely the diabetic diet and the low-sugar diet, did not meet this goal. Instead, they exceed the recommended fat intake (of 35% of total energy at maximum), with 36.34% in the diabetic diet and 38.08% in the low-sugar diet. While of great interest *per sé*, both macronutrients are not directly related to DAL calculations (see Section 2.4).

A typical Western diet is associated with an acid load from the diet ranging between approximately 50 and 75 mEq/d [38,39]. The mean NEAP_F_ in our study was also in that rangem and we thus present preliminary evidence that the examined special diets are not associated with a clinically relevant reduction in DAL scores. In comparison, it is noteworthy that fully plant-based vegetarian and vegan diets have been shown to substantially decrease DAL scores. Kahleova et al. examined the effects of a low-fat vegan diet (unrestricted in calories) on DAL [6]. After just 4 months, median PRAL_R_ values fell from 3.6 (0.4 to 6.8) to −20.7 (−23.3 to −18.1) mEq/d, and median NEAP_F_ values fell from 50.8 (47.1 to 54.5) to 25.7 (24.0 to 27.4). A 2017 study by Cosgrove and Johnston demonstrated a comparable picture: after seven consecutive days on a (strict) vegan diet, mean PRAL_R_ values fell from 23.7 ± 17.7 to −6.0 ± 12.8 [19].

To the best of our knowledge, this is the first study using nationally representative data from the NHANES to investigate DAL of popular special diets in the United States. Our findings are of high translational value, given that physicians and other healthcare providers often prescribe specific diets for various purposes (e.g., a high-fiber diet for constipation or a weight-loss diet to lower body weight). None of the examined diets was associated with lower DAL scores than those found in a Western diet, a fact that has to be kept in mind when tailoring a diet to an individual’s needs and disease profile.

### Strengths and Limitations

Our analysis has several limitations that warrant further discussion. As described earlier, dietary patterns were self-reported, and no specific list of diets or standardized definitions were provided. Participants described their diets, and their responses were categorized by the specifically trained NHANES interviewers. Although NHANES staff are highly experienced and trained in nutritional interviews, this may introduce a certain bias. As this is a secondary data analysis of available data, we may not report additional details from the interviews. Moreover, the cross-sectional nature of our data is an intrinsic limitation that does not allow for causal interference.

An additional limitation is the lack of a food group analysis, which would have enriched this investigation. However, in order to maintain the large sample size, we refrained from this step. A food group analysis would have helped to better distinguish the special diets presented here, particularly with regard to the lack of publicly available definitions (see above). On the other hand, we present reliable nutrient intake data from the NHANES, which is most important when calculating DAL scores. It is conceivable that the dietary patterns differed with regard to their food intake (particularly when looking at macronutrient distributions); however, for the DAL calculations themselves, only micronutrients and total protein intake were required. We refer the interested reader to the work of Remer and Manz, which presents PRAL values from selected foods in detail [30].

Finally, we acknowledge that this analysis included only two DAL scores (PRAL_R_ and NEAP_F_). Another common score, the NEAP score by Remer et al. [30], which also considers anthropometry-based estimates for organic acid excretion, was not included in this study. This score relies on reliable anthropometric data, and adding it would have further decreased the sample size. Due to the very limited number of comparable studies, comparison to other trials also remains difficult. Thus, our study has a rather exploratory character, and future studies will be required to confirm our findings.

On the other hand, this study draws upon a number of strengths. It is based on a nationally representative and large dataset (National Health and Nutrition Examination Survey) representing the civilian non-institutionalized US population. Nutrient and energy intake data were obtained from the NHANES dietary interview component, a module using a computerized 24-h dietary recall method to estimate energy and nutrient intake for all participants [28]. The employed dietary data collection instrument (the Automated Multiple Pass Method (AMPM)) is a validated tool and has been used in numerous studies. Additional information on the employed instrument can be obtained from the homepage of the U.S. Department of Agriculture [42]. Third, and probably most important, our study may serve as an incentive for others to explore in greater detail the potential benefits and health repercussions of special diets (other than plant-based diets) with regard to DAL.

To the best of our knowledge, this is one of the first studies to specifically explore the DAL of special diets using nationally representative dietary data from the NHANES. Future studies should seek a better description and definition of the examined special diets (e.g., supplemented by food group analyses and other nutrient intake data unrelated to DAL calculations). Importantly, future studies should also include all three DAL scores and anthropometric data (see limitations).

## 5. Conclusions

Our data suggest that none of the examined popular special diets in the NHANES (weight-loss diet, low-fat diet, low-salt diet, low-sugar diet, diabetic diet) were associated with negative DAL scores. The estimated DAL scores of the examined diets were altogether positive and comparable to Western diets, indicating acidifying properties. In comparison to vegetarian and vegan diets, which usually yield negative or near-negative total PRAL_R_ values, the special diets examined here were associated with PRAL_R_ scores > 9 mEq/d, which is indicative of acidifying properties. The estimated NEAP_F_ scores of special diets were comparable with the ones of Western diets in previous studies. Additional studies confirming our results are warranted. In the meantime, it might be advisable to rely on established DAL-lowering strategies (e.g., by increasing the intake of vegetables and fruits and by avoiding or reducing processed meat products abundant in sulfur-containing amino acids and preservative phosphate).

## Figures and Tables

**Figure 1 ijerph-19-05748-f001:**
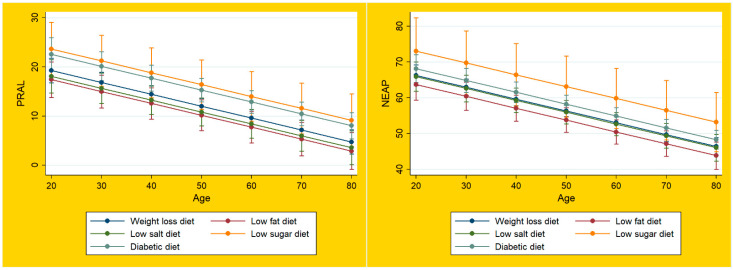
Dietary acid load of special diets in the NHANES. Legend for Figure 1: Plots of marginal predicted values based on the regression model, illustrating differences in the linear relationship of PRAL_R_ and NEAP_F_ and age, depending on each special diet.

**Table 1 ijerph-19-05748-t001:** Demographic data and nutrient and energy intake in individuals reporting consumption of a special diet.

	Weight Loss Diet	Low Fat Diet	Low Salt Diet	Low Sugar Diet	Diabetic Diet	*p*
Demographic data						
**Sex**						0.004 ^c^
Male	*n* = 720 (36.02 (1.53))	*n* = 172 (48.25 (4.05))	*n* = 190 (45.84 (3.38))	*n* = 65 (41.93 (5.52))	*n* = 315 (43.44 (2.63))	
Female	*n* = 1166 (63.98 (1.53))	*n* = 194 (51.75 (1.53))	*n* = 234 (54.16 (1.53))	*n* = 83 (58.07 (1.53))	*n* = 413 (56.56 (1.53))	
**Age**	45.91 (0.70)	49.99 (1.14)	59.46 (1.04)	51.63 (2.07)	60.90 (0.78)	<0.001 ^d^
**Race/Ethnicity**						0.003 ^c^
Mexican American	*n* = 263 (7.80 (1.04))	*n* = 73 (13.04 (2.58))	*n* = 50 (6.69 (1.36))	*n* = 14 (4.46 (1.67)) ^b^	*n* = 125 (9.22 (1.62))	
Other Hispanic	*n* = 204 (6.13 (0.80))	*n* = 50 (6.86 (1.35))	*n* = 44 (6.15 (1.82))	*n* = 19 (5.42 (2.10)) ^b^	*n* = 88 (5.40 (0.83))	
Non-Hispanic White	*n* = 761 (67.59 (1.77))	*n* = 130 (60.39 (3.86))	*n* = 155 (61.04 (3.56))	*n* = 71 (75.63 (3.58))	*n* = 291 (66.26 (2.60))	
Non-Hispanic Black	*n* = 403 (10.35 (0.84))	*n* = 64 (8.74 (1.54))	*n* = 123 (18.36 (2.20))	*n* = 22 (6.44 (1.76)) ^b^	*n* = 153 (11.44 (1.40))	
Other Race ^a^	*n* = 255 (8.12 (0.70))	*n* = 49 (10.97 (2.84))	*n* = 52 (7.76 (1.62))	*n* = 22 (8.03 (1.93))	*n* = 71 (7.68 (1.10))	
Nutrient and energy intake						
**Energy** (kcal)/d	1910.85 (30.38)	1960.92 (61.98)	1946.78 (62.48)	1830.74 (92.19)	1821.81 (47.14)	0.289 ^d^
**Protein** (g/d)	79.64 (1.59)	80.03 (3.14)	77.40 (3.03)	84.60 (4.36)	77.37 (2.21)	0.619 ^d^
**Protein** (%/te)	17.28 (0.22)	16.93(0.47)	16.34 (0.37)	18.96 (0.70)	17.51 (0.31)	0.012 ^d^
**Carbohydrate** (g/d)	218.20 (3.82)	236.38 (8.16)	241.69 (7.50)	190.46 (10.65)	208.08 (5.74)	<0.001 ^d^
**Carbohydrate** (%/te)	46.40 (0.45)	48.66 (0.81)	50.73 (0.96)	42.44 (1.59)	42.44 (1.59)	<0.001 ^d^
**Fat** (g/d)	75.92 (1.59)	73.9 8 (2.80)	71.81 (3.34)	79.40 (4.52)	75.47 (2.23)	0.649 ^d^
**Fat** (%/te)	34.77 (0.41)	33.28 (0.70)	32.18 (0.72)	38.08 (1.01)	36.34 (0.39)	<0.001 ^d^
**Calcium** (mg/d)	950.01 (17.84)	975.58 (47.72)	895.45 (42.12)	1110.89 (87.87)	936.70 (9.69)	0.195 ^d^
**Magnesium** (mg/d)	304.72 (6.16)	326.07 (13.06)	296.580 (10.35)	326.88 (18.24)	290.03 (7.80)	0.122 ^d^
**Phosphorus** (mg/d)	1334.36 (24.18)	1366.06 (54.62)	1307.94 (48.72)	1410.43 (76.83)	1329.55 (35.11)	0.804 ^d^
**Potassium** (mg/d)	2625.65 (46.72)	2732.91 (101.59)	2742.61 (107.63)	2640.02 (131.84)	2629.62 (61.12)	0.718 ^d^
**Sodium** (mg/d)	3327.03 (59.90)	3370.09 (129.93)	3333.89 (127.15)	3285.39 (178.36)	3287.34 (84.99)	0.987 ^d^

Table 1 legend: ^a^ = including Multi-Racial, ^b^ = unreliable estimated proportion as per NCHS guidelines, ^c^ = based on Stata’s design-adjusted Rao–Scott test, ^d^ = based on regression analyses followed by adjusted Wald tests. Abbreviations: te = total energy.

**Table 2 ijerph-19-05748-t002:** DAL scores in individuals reporting consumption of a special diet.

	Weight-Loss Diet	Low-Fat Diet	Low-Salt Diet	Low-Sugar Diet	Diabetic Diet	*p*
**PRAL**_R_ (mEq/d)	12.98 (0.94)	11.21 (1.70)	9.37 (1.58)	15.26 (2.88)	12.17 (1.35)	0.328 ^a^
**NEAP**_R_ (mEq/d)	57.50 (0.95)	54.08 (1.75)	53.45 (1.68)	62.05 (4.54)	54.51 (1.30)	0.052 ^a^

Table 2 legend: ^a^ = based on regression analyses followed by adjusted Wald tests.

**Table 3 ijerph-19-05748-t003:** Linear regression models investigating associations of special diets and (1) PRAL_R_ and (2) NEAP_R_ scores.

	PRAL_R_	*p*	NEAP_F_	*p*
**Gender**				
Female	−3.66 (−5.87–(−1.46))	0.001	−2.35 (−4.57–(−0.12))	0.039
Male	-		-	
**Age** (years)	−0.24 (−0.30–(−0.18))	<0.001	−0.33 (−0.40–(−0.26))	<0.001
**Ethnicity**				
Mexican American	3.62 (−0.15–7.38)	0.059	2.38 (−0.62–5.38)	0.118
Other Hispanic	−2.02 (−5.73–1.70)	0.283	0.43 (−3.51–4.37)	0.830
Non-Hispanic White	-		-	
Non-Hispanic Black	2.35 (0.19–4.52)	0.033	6.19 (3.28–9.10)	<0.001
Other Race	0.15 (−3.47–3.76)	0.936	1.72 (−1.50–4.93)	0.291
**Energy intake** (kcal/d)	0.009 (0.008–0.011)	<0.001	0.002 (0.001–0.004)	
**Special diet**				
Weight-loss diet	1.18 (−2.21–4.57)	0.490	0.34 (−3.18–3.86)	0.846
Low-fat diet	−0.67 (−5.08–3.74)	0.763	−2.20 (−7.02–2.62)	0.366
Low-salt diet	-		-	
Low-sugar diet	5.58 (−0.43–11.59)	0.069	7.15 (−1.73–16.03)	0.113
Diabetic diet	4.47 (1.00–7.94)	0.012	2.25 (−1.49–5.99)	0.234

Table 3 legend: Coefficients are displayed with their 95% confidence intervals and *p*-values. The symbol “-” indicates the reference category. *p* = *p*-value.

## Data Availability

Data are publicly available online (https://wwwn.cdc.gov/nchs/nhanes/Default.aspx; accessed on 4 April 2022). The datasets used and analyzed during the current study are available from the corresponding author upon reasonable request.

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
