# Peer review of "Nutrient Intake and Dietary Acid Load of Special Diets in the NHANES: A Descriptive Analysis (2009–2018)"

_ijerph, 2022, doi:10.3390/ijerph19095748_

Round 1
Reviewer 1 Report
In this research authors quantified dietary acid load (DAL) and nutrient intake in some of the most popular special diets in the United States in order to estimate their health effect. Special diets that were analyzed: (1) weight loss/low calorie diet, (2) low fat/low cholesterol diet, (3) low salt/low sodium diet, (4) sugar free/low sugar diet, and (5) diabetic diet. Based on the obtained results for all examined diets, DAL scores were consistently positive, indicating their acidifying properties. None of the special diets had an alkaline character.
Topic is relevant because DAL is very important and tells about potential health impact considering the dietary acids and their impact on health. Authors did made a good choice to investigate DAL in most popular diets in US giving more information about healthiness of these diets.
DAL as a novel nutritional target was analyzed for its connection with some metabolic and health problems (diabetes, kidney problems, some cancers, migraine, etc.) and in the different types of diet (Western, Eastern, Vegetarian) and this research adds new information for DAL levels in most popular diets in US shedding more light in this area of research.
The methodology in this research is fine and authors themselves correctly addressed its limitations in their Discussion section (under 4.1 Strength and limitations).
The conclusions are appropriate and based on obtained data and address well the main question.
The references are appropriate and sufficient.
There are needed some minor corrections:
Line 41 : metabolic... it should be metabolic syndrome??
Line 143: in Table 1. Please align Race/Ethnicity with data columns
Author Response
Dear Reviewer,
Please find our detailed point-by-point response below.
Sincerely,
The authors

Reviewer 2 Report
The research work Nutrient Intake and Dietary Acid Load of Special Diets in the NHANES A Descriptive Analysis (2009-2018) is very interesting, I only have minimal suggestions to improve this manuscript:
In the introduction section of the manuscript (Page 1; lines 34-36 it is mentioned
“Meat and dairy products are abundant in sulfur containing amino acids (cysteine, homocysteine and methionine) and preservative phosphate, and thus have an acidifying effect on the human body (Figure 1) [2,3].”
In the interest of a better understanding of this problem, the authors could add more specific information about the mechanism at the cellular level that causes these foods to acidify the cells of the human body. Or more specific information could also be added on why Plant-based diets (PBD) have alkaline properties.
Page 3; lines 109-124:
Los autores mencionan:
“For the statistical analysis, we used Stata version 14 (StataCorp., College Stadion, TX, USA) and constructed appropriate………………….”
In scientific articles it is suggested to report the information in the third person, for which "we used" (first person), “we described”, “we followed”, “Our análisis”, could be substituted.
Author Response

(The authors gave the same response as above.)

Reviewer 3 Report
The article "Nutrient Intake and Dietary Acid Load of Special Diets in the 2 NHANES: A Descriptive Analysis (2009-2018)" describes the role of dietary acid loads on western (US) diets based on a NHANES survey method. I am requesting the authors to thoroughly restructure the article as it is difficult to comprehend a clear workflow. I have the following comments for the article that can be considered:
1) I am unable to understand the study design from Materials and methods section. Was this study conducted in the US population? If so, a clear workflow pertinent to section 2.2 (lines 79-89) must be presented, along with an ethical approval (if any applicable).
2)It is absolutely essential to elucidate the pattern of questionnaire and the diets (according to the NHANES interview or modified) - in the introduction section of this manuscript. The actual questionnaire can be added as supplementary material. The authors cite their previous work on vegan diet and its relevance to DAL, however, the current article reports results from NHANES interviews diet. It is necessary to distinguish the plant based sources and the animal-based sources in the respective diets (if any data is available).
2) Figure 2: I suggest establishing statistical significance between the different plots. It would be interesting to see where these plots stand from a baseline calculation from any previous published data for vegan diets (if data is available from literature).
3)Section 4.1 and section 5: I am finding it very difficult to agree with the conclusions that the authors have made. While I do acknowledge the study does recommend exploring more of PBD with respect to DAL in western diets, I urge the authors to strengthen their arguments by atleast considering how the 5 diets under experiment have distinguishable plant based and animal based sources. There is not enough evidence currently supporting the claims - Kindly modify the text to address the comments to make the arguments stronger. Discussions pertinent to age and racial demography are also very important pertinent to the discussion.
4) Minor: Fig 1 depicts citrus fruits/grapes as alkaline foods. Is this correct?
Author Response

(The authors gave the same response as above.)

Reviewer 4 Report
- The abstract can be fine-tuned
- Line 33, the (saturated) should shift after fat
- Line 35, what are the preservative phosphates?
- Line 37, did all the fruits acidic? Give specific information
- Line 41, check the sentence is incomplete
- Line 47, After Figure 1. remove the " word"
- I think lines 48, 49, and 50 should remove
- Try to give the existed work and the Gap related to this area.
- Need to work on the introduction
- Line 75 and 76. Is this the 24 hours’ recall method?
- Line 83 to 87, give standard definitions to the diets
- Line 124 and 125, started with the methodology, so move this to the materials and method section
- Line 143, remove the word " title"
- in table 1, the first rows used n many times better to use once and mention in the heading or in any appropriate place
- In the second part of the table give values in the parenthesis what do they represent?
- Line 144, Table 1 legend
- Line 162, remove "title"
- Remove lines 163- 165
- Discussion is very narrow, better discuss in multi-direction aspects of the diets
- Conclusions are superficial, better give little deep
Author Response

(The authors gave the same response as above.)

Reviewer 5 Report
Dear Authors,
I like the idea of using the NHANES findings to measure DAL among individuals following different diets. While reading the manuscript, I got the impression of too little attentiveness on the part of the Authors, and maybe even a rush. On the other hand, I appreciate the smaller length of the manuscript compared to others, but I have some suggestions to increase the scientific value of the manuscript.
In the Introduction, I suggest that citations concerning the obvious and well-known features of the Western diet be dropped, and that the focus be on the health effects of DAL. This has been too briefly presented (only L.38-43). It would be worth writing about why adequate potassium intake is important, what cardiovascular diseases are associated with DAL, how DAL affects insulin resistance, cortisol secretion, excretion of calcium and magnesium, carbohydrate metabolism and lipid profile, among others. The facts from the first sentence in the discussion should be here, but in more detail. I also suggest to drop Figure 1, which serves no purpose and is certainly not an illustration of the second sentence from the Introduction. Besides, according to the latest knowledge, vegetables should be consumed in the largest quantities, whereas in the figure fruit is favoured. The formulas for the NEAP and PRAL indices, fundamental to the issue under study, should be included in subsection 2.3.
The Introduction and Discussion give the impression that the authors equate the term plan-based diet with the term vegetarian or vegan diet. However, the term PBD also refers to the flexitarian diet, the DASH diet, the Mediterranean diet or the sustainable diet. Therefore, I suggest using precise terms (e.g. L.51-54). Today, when human activity has led to the crossing of several planetary boundaries that define a safe space for humanity in the Earth system, it is difficult to agree with the authors' approach that "not everyone wishes to adopt a PBD" (a phrase used twice).
The aim should be more precisely formulated - please indicate how many NHANES studies were used or from which period and which diets were taken into account.
I suggest enlarging Table 1 by two rows and recording the results of PRAL calculations in one and NEAP in the other. This will certainly improve the perception of the most important results in the manuscript. Also, the Results section only shows DAL by age, why are there no graphs by the other variables mentioned?
The Discussion omits at all the assessment of nutrients and energy intake, the first issue included in the title of the manuscript. How do the authors assess dietary energy structure, macro- and micronutrient intake compared to dietary recommendations? In assessing DAL indicators, I suggest quoting PRAL and NEAP values from other studies, show the range of magnitudes based on the literature review. The perfunctory information (L. 174, 182) is unsatisfactory. What are these indicators for different plant-based diets, for a vegan diet, different vegetarian diet options?
The authors indicated in the limitations of the study that they deliberately omitted to analyse food groups due to the large number of individuals surveyed. Thus, I conclude that the results of the NHANES study show 24-hour intake data also in the form of food products. I therefore suggest that the authors supplement their paper by characterising the composition of the diets analysed. Some markers could be chosen, e.g. the share of vegetables and meat (including their products) in the weight of daily intake or the intake of plant and animal protein, etc. The argument in favour is also the first limitation pointed out by the authors. It is reflected in the higher sodium intake of people on a low salt diet than those on a weight loss, low sugar and diabetic diets.
Minor comments:
L.41 - 'disorders' missing after 'metabolic'
L.102 and 105 - the full formula names should be written in the same way, either in upper or lower case.
For the whole text, starting from L.102 - adopt one way of writing - either PRALR and NEAPF, or no subscript letters.
L.143-144 - remove the word 'title:' from the table title and move up one line the data for race. Write the table legend in a smaller font and align under the table (i.e., this should not be a section of text).
L.162 - remove the word 'title:' from the figure title, the explanations to the figure (legend) should be placed above the figure title and should be written in smaller font.
L.184 - probably by mistake the authors here quote source 21, which refers to the KNHANES study (in Korea).
Kind regards.
Author Response

(The authors gave the same response as above.)

Round 2
Reviewer 3 Report
I thank the authors for careful consideration and modification of the manuscript and satisfactory responses to all the concerns. Further I have a few minor comments for the manuscript that can be considered.
1) Please consider modifying/removing Fig 1. It can be graphical abstract. But in the main manuscript the relevance is not justified.
2) Regarding Table 3, I am not certain if a score comparison is possible for PRAL or NEAP between gender/ethnicity groups as the initial sample sizes (n numbers) were statistically significantly different. Kindly check.
3) Regarding Table 2: It is quite remarkable why PRAL value between diets is not significant but NEAP is. Since both parameters include total protein (significantly different between special diets) but the other parameters (P, K, Mg or Ca) are not significantly changing. I am requesting an explanation for this phenomenon.
4) Discussion lines 315 onwards: Reference to vegetarian diets are made here onwards, but current analysis is only based on special diets from NHANES. It would be really nice if a reference supplemental table is provided for a baseline (plant based/vegetarian/vegan) diet for the relevant elements to improve impact of the current findings.
Author Response
Dear Reviewer,
Please find our point-by-point response attached.
Sincerely,
The authors

Reviewer 5 Report
Dear Authors,
Thank you for your point by point response to the suggestions in my review. I am satisfied with the amendments made to the manuscript, which have significantly increased its scientific value. Of course, some of my suggestions were not taken on board, but the authors have detailed their point of view, which fully justified their position. It will be my pleasure and satisfaction to recommend the publication of the article to the editors. While reading the new version of the manuscript, I still noticed minor technical errors (L. 81 - should be written 'significantly', L. 173 - the redirect to Figure 1 should be removed, L.322 - the opening bracket sign is missing).
Kind regards.
Author Response

(The authors gave the same response as above.)
